# *Leptodactylus latrans* Amphibian Skin Secretions as a Novel Source for the Isolation of Antibacterial Peptides

**DOI:** 10.3390/molecules23112943

**Published:** 2018-11-11

**Authors:** Alvaro Siano, Maria Veronica Humpola, Eliandre de Oliveira, Fernando Albericio, Arturo C. Simonetta, Rafael Lajmanovich, Georgina G. Tonarelli

**Affiliations:** 1Departamento de Química Orgánica, Facultad de Bioquímica y Cs. Biológicas (FBCB), Universidad Nacional del Litoral (UNL), Ciudad Universitaria, 3000 Santa Fe, Argentina; mvhumpola@gmail.com (M.V.H.); tonareli@fbcb.unl.edu.ar (G.G.T.); 2Consejo Nacional de Investigaciones Científicas y Técnicas (CONICET), 1825 Buenos Aires, Argentina; lajmanovich@hotmail.com; 3Proteomics Platform, Barcelona Science Park, Baldiri Reixac 10, 08028 Barcelona, Spain; eoliveira@pcb.ub.es; 4CIBER-BBN, Networking Centre on Bioengineering, Biomaterials and Nanomedicine, Barcelona Science Park, Baldiri Reixac 10, 08028 Barcelona, Spain; 5Department of Organic Chemistry, University of Barcelona, 08028 Barcelona, Spain; 6School of Chemistry and Physics, University of KwaZulu-Natal, 4000 Durban, South Africa; 7Cátedras de Microbiología y Biotecnología, Departamento de Ingeniería en Alimentos, Facultad de Ingeniería Química, U.N.L. Santiago del Estero 2829, 3000 Santa Fe, Argentina; asimonet@fiq.unl.edu.ar; 8Cátedra de Ecotoxicología, Escuela Superior de Sanidad. FBCB, U.N.L. Ciudad Universitaria, 3000 Santa Fe, Argentina

**Keywords:** peptides, frogs, synthesis, antimicrobial, peptidomics

## Abstract

Amphibians´ skin produces a diverse array of antimicrobial peptides that play a crucial role as the first line of defense against microbial invasion. Despite the immense richness of wild amphibians in Argentina, current knowledge about the presence of peptides with antimicrobial properties is limited to a only few species. Here we used LC-MS-MS to identify antimicrobial peptides with masses ranging from 1000 to 4000 Da from samples of skin secretions of *Leptodactylus latrans* (Anura: Leptodactylidae). Three novel amino acid sequences were selected for chemical synthesis and further studies. The three synthetic peptides, named P1-Ll-1577, P2-Ll-1298, and P3-Ll-2085, inhibited the growth of two ATCC strains, namely *Escherichia coli* and *Staphylococcus aureus.* P3-Ll-2085 was the most active peptide. In the presence of trifluoroethanol (TFE) and anionic liposomes, it adopted an amphipathic α-helical structure. P2-Ll-1298 showed slightly lower activity than P3-Ll-2085. Comparison of the MIC values of these two peptides revealed that the addition of seven amino acid residues (GLLDFLK) on the N-terminal of P2-Ll-1298 significantly improved activity against both strains. P1-Ll-1577, which remarkably is an anionic peptide, showed interesting antimicrobial activity against *E. coli* and *S. aureus* strain, showing marked membrane selectivity and non-hemolysis. Due to this, P1-L1-1577 emerges as a potential candidate for the development of new antibacterial drugs.

## 1. Introduction

The rise of bacterial resistance to conventional antibiotics is one of the major causes of high mortality rates and inefficient therapy. Regarding this, intensive research efforts are being channeled into the development of new antimicrobials agents [1]. Traditionally, the principal targets of antibacterial drugs have been the bacterial cell walls and protein synthesis [2,3].

Regardless of the emerging technologies used for drug discovery, Nature continues to be one of the most important sources of molecules for the development of novel therapeutic agents [4,5]. Only 20% of marketed compounds with antimicrobial activity are synthetic and not inspired by Nature [6,7]. 

The relevance of peptides in drug discovery programs has recently increased [8,9,10]. Antimicrobial peptides (AMPs) are produced by a variety of living organisms, including fungi, bacteria, plants, vertebrates, and invertebrates. These natural molecules are involved in innate immunity and form the first line of defense against pathogens [11,12]. In animals, AMPs supplement the humoral and cellular immune system of the host and are produced in the skin, epithelial tissues, and acute inflammatory cells [13,14,15]. In general, AMPs contain a significant proportion of hydrophobic amino acid residues and have a positive net charge (that varies from +2 to +9).

Historically, cationic AMPs were believed to act only by disrupting the integrity of the bacterial membrane. Bacterial membranes are formed primarily of anionic lipids and do not contain cholesterol, while mammalian membranes comprise mainly neutral lipids and contain cholesterol [16]. In this sense, cationic AMPs are electrostatically attracted to bacterial membranes, and the hydrophobic amino acid residues facilitate interactions with fatty acyl chains.

Several studies have demonstrated that peptides can translocate across the bacterial cytoplasmatic membrane to inhibit multiple internal targets, including protein folding, cell wall and DNA/RNA synthesis, translocation and cell division [17,18,19].

The skin of amphibian serves as a defense system, producing AMPs that are effective against a wide range of pathogenic microorganisms and predators [16,17,18]. Granular glands (also called serous glands or poison glands) produce a large variety of such bioactive substances, including AMPs [16,20], neurotoxic peptides [21], gastric disturbance peptides [14] and alkaloids [22]. The center of the gland is filled with granules containing active peptides [23], and when the animal is injured or alarmed, the content is released through skin secretions [24,25].

Leptodactylus is a genus of frog that includes more than 60 species that have a geographical distribution ranging from southern North America to South America. *Leptodactylus latrans* (formerly known as *L. ocellatus*) is a common species of the family Leptodactylidae in the Neotropical region.

Here we report the purification of two antimicrobial peptides isolated from *L. latrans*, together with their synthesis and that of a hybrid peptide based on a previously reported occelatin. We also report on the antimicrobial and hemolytic properties of these three synthetic peptides.

## 2. Results

In a previous study, we examined two widely used extraction methodologies [transcutaneous amphibian stimulation (TAS) and solvent extraction (SE)]. Both methods rendered material that inhibited the growth of bacterial strains, against a wide range of bacteria, including *Mycobacterium tuberculosis*, *Escherichia coli*, *Bacillus cereus*, *Staphylococcus aureus* and *Pseudomonas spp.* strains [26]. Here we applied mass spectrometry (MS-MS and LC-MS-MS), in the molecular weight range from 1000 to 4000 Da, in an extract of *L. latrans* skin secretion obtained by TAS, to identify antimicrobial peptides.

### 2.1. Analysis and Purification of the Complete Extract of L. latrans

Figure 1 shows the chromatogram of the dialyzed fraction of *L. latrans* and the fractions that were separated by HPLC. The six fractions were tested against *S. aureus* ATCC 25923 and *E. coli* ATCC 25922 using the agar diffusion assay. Fractions 1, 2 and 5 inhibited the growth of the Gram (−) strain, while only Fraction 4 inhibited the Gram (+) and Gram (−) strains. We, therefore, selected Fraction 4 for further studies.

### 2.2. Mass Spectrometry Analysis of Fraction 4 of L. latrans

Fraction 4 was separated, desalted, and directly analyzed by MS without any further purification. The amino acid sequences of some peptides were inferred through “de novo” analysis using the Peaks Studio software. Only those with a “de novo” peak scores over 85% were considered. 

The mass spectrum of Fraction 4 is shown in Figure 2 and the identified amino acid sequences are listed in Table 1 (the MS-MS spectra are shown in the Appendix A).

The amino acid residues Leu/Ile and Gln/Lys, which could not be distinguished by MS, were inferred on the basis of similarity with homologous Ranidae amino acid sequences whenever possible [27,28,29,30]. According to the predictive analyses of the secondary structure, only the amino acid sequence AAGKGLVSNLLEK showed contributions of α-helix and the amino acid sequence DEMKLDGFNMHLE corresponded to an unstructured anionic peptide. For the other amino acid sequences, the predictive analysis could not be considered by GOR V or PSIPRED methods because of the short length of the peptide.

The most cationic and hydrophobic amino acid sequence, S1, was analyzed through the AMP database APD (http://aps.unmc.edu/AP). The results of the alignment allowed us to determine the most similar peptides in the database (Appendix A) and revealed that S1 shows significant similarity to the 8–17 region of occelatin 5 [31].

### 2.3. Solid-Phase Peptide Synthesis

The following three novel peptides were synthesized (see Table 2), and subsequent studies of their structure and antimicrobial activity were performed: 

1. Peptides corresponding to two identified amino acid sequences from *L. latrans* (ID: S1 and S2) corresponding to the [M + H]^+^ = 1578.71 and [M + H]^+^ = 1299.72. ID: P1-Ll-1577 and P2-Ll-1298, respectively. 

2. A hybrid peptide consisting of the combination of two fragments: the region 1–7 of ocellatin 5 (GLLDFLK) reported in UniProt database (Entry name: OCE5_LEPOE) as N-terminal, followed by P2 -Ll-1298 (Table 2). ID: P3-Ll-2085.

### 2.4. Antimicrobial and Hemolytic Activity of the Synthetic Peptides

The MIC values obtained for the three analogs against *E. coli* ATCC 25922 and *S. aureus* ATCC 25923 are shown in Table 3.

All peptides, except P1-Ll-1577, were dissolved in 36 µL of Milli-Q water and subsequently diluted to the maximum concentration tested (1280 µg/mL). The anionic peptide identified as P1-L-1577 (net charge: −2) was dissolved in a buffer of zinc chloride and sodium chloride. In the absence of these ions, the peptide showed no inhibitory activity against any of the strains. In many cases, the presence of cations (Zn^+2^, Na^+^, Mg^+2^, etc.) led to the interaction of anionic peptides with the bacterial membrane, as reported elsewhere [32].

The MIC values of P1-Ll-1577 are shown in Table 3. This peptide showed significant inhibitory activity against both strains, but in particular against *E. coli*. P3-Ll-2085 presented a MIC of 15 µM against both bacterial strains while P2-Ll-1298 showed slightly lower activity, presenting a MIC of 24.6 µM against *E. coli* and 49 µM against *S. aureus*. Comparison of the MIC values of the two analogs revealed that the addition of seven amino acid residues (GLLDFLK) on the N-terminal of P2-Ll-1298 significantly improved inhibitory capacity.

Figure 3 shows the hemolytic activity of the analogs. At the MIC value, none of the peptides, except P3-Ll-2085, presented percentages of hemolysis above 10%. At the highest tested concentration (320 µM), P2-Ll-1298 reached the 100% hemolysis while P1-Ll-1577 caused only 12%.

At 40 µM, P3-Ll-2085 caused 100% hemolysis; This observation is consistent with the high hydrophobicity (H: −0.01) and amphipathicity (µH: 0.447) of the amino acid sequence, calculated using the Eisenberg scale [33]. Studies with helical peptides have shown that an increase in cationicity enhances antimicrobial activity while an increase in hydrophobicity, amphipathicity and helicity favor hemolytic activity and loss of selectivity for microorganism membranes [34].

All the analogs showed a therapeutic index (TI) over 1 (Table 4), meaning that total hemolysis is not achieved at the MIC. P1-Ll-1577 showed exciting TI values, especially for *E. coli*. The lowest TI value was obtained with P3-Ll-2085, comparable with that of toxins such as melitin, for which TI values of 0.6 against Gram (+) and Gram (−) bacteria have been reported [35].

### 2.5. Secondary Structure Determination by Circular Dichroism (CD)

The CD spectra of the analogs are shown in Figure 4. Spectra were collected in four distinct environments: water, trifluoroethanol (TFE)/water (50% *v*/*v*), and in the presence of DPPG or DPPC vesicles.

The CD spectra showed that none of the compounds adopted a preferential conformation in water, which is consistent with the presence of a minimum at 198 nm. In the presence of TFE, P3-Ll-2085 adopted a helical conformation, an observation consistent with the presence of two minimums at 205–207 nm and 215–220 nm and a maximum at 195 nm.

Deconvolution spectra by SELCON and CONTILL methods indicated more than 70% helical structure for P3-Ll-2085. P2-Ll-1298 also showed contributions of α-helix, while P1-Ll-1577, although presenting contributions of turn structure, was less ordered in the presence of TFE (percentage of an unordered structure higher than 40%).

In the presence of DPPG vesicles (Figure 4C), P3-Ll-2085 and P2-Ll-1298 adopted a helical structure. Nevertheless, the higher molar ellipticity values registered for the former, together with slight shifts of the positions of the minima, indicate further stabilization of the helix about the latter. The spectra deconvolution of P3-Ll-2085 and P2-Ll-1298 by SELCON 3 and CONTILL showed contributions of α-helix of over 70% and 50%, respectively.

P1-Ll-1577 did not interact with DPPG vesicles, an observation that can be explained by its anionic character (net charge of −2).

Most of the peptides were not ordered in the presence of DPPC vesicles. Nevertheless, for P3-Ll-2085, data deconvolution by CONTILL indicated partial contribution of α-helix.

In spite of the wide variety of the amino acid sequences found in amphibian AMPs, several reports have revealed that, in general, they tend to form amphipathic α-helical structures in the presence of membrane-mimetic micelles, liposomes, and organic solvent mixtures. Of the three synthesized peptides here, only two showed α-helix contributions.

## 3. Discussion

Data in the literature on the antimicrobial activity of peptides isolated from amphibians vary greatly. Magainin 2 (isolated from *Xenopus laevis*) has a MIC of 100 µM against *S. aureus* and *E. coli* [36]. Many peptides with activity against Gram (+) and Gram (−) bacteria in a range of concentrations that vary from 40 µM to 100 µM have been isolated from the Leptodactylidae family [37,38,39,40,41,42]. Among them, six ocellatins isolated from *L. latrans*, including ocellatin 4, which has been the most widely studied, inhibit strains of *E. coli* and *S. aureus* with MIC values of 64 µM [40].

Several peptides isolated from the Hylidae and Leptodactylidae families show low hemolytic activity. These include hylins (isolated from *H. biobeba*) [43,44], and leptoglycin and pentadactylin (*L. pentadactylus*) [38,42], among others. In contrast, ocellatin 4 produces 50% hemolysis at 14.3 µM [40].

The therapeutic potential of many temporins is limited due to their high hemolytic activity against human erythrocytes, except temporin A and B, which show scarce hemolytic activity [45,46].

A direct correlation between cytolytic activity and hydrophobicity have been demonstrated with studies with model α-helical peptides. Other parameters, such as hydrophobic moment, amphipathicity, helicity, and the size of the hydrophobic/hydrophilic domain influence selectivity and membrane interaction [16,47,48]. 

Although there is no consensus about the precise mechanism underlying the action of AMPs, it is acknowledged that cytolytic activity is not mediated through the interaction with specific receptors. Several reports have shown that peptide-lipid interactions lead to membrane permeation and that these interactions play a vital role in the activity of AMPs. Membrane permeation by amphipathic helical peptides has been suggested to occur via one of two mechanisms, namely transmembrane pore formation via a “barrel-stave” mechanism or membrane solubilization/ destruction via a “carpet” mechanism [16,49,50,51,52,53,54,55].

The “barrel-stave” model implies the formation of transmembrane pores/channels by bundles of amphipathic helices, such that the hydrophobic surfaces interact with the lipid core of the membrane and their hydrophilic surfaces point inward, producing an aqueous pore. Peptides with cytolytic properties toward both mammalian cell membranes and bacteria are considered to act following the “barrel-stave” model [52].

In a membrane-mimetic environment, temporins have the propensity to form a stable amphipathic α-helix. The lack of selectivity of these peptides may be related to the low positive net charge of the peptides, and the hydrophobic interactions are responsible for the binding to the membrane. In this regard, Mangoni et al. [50] proposed that temporins use a “barrel-stave” mechanism. It has also been reported that the peptide Ctx-Ha isolated from *Hypsiboas albopunctatus*, act by this mechanism [54].

In the “carpet” model proposed by Shai and co-workers [52,56], the membrane is permeabilized in a “detergent-like” manner. The peptides are in contact with the lipid head groups during membrane permeation, and they do not insert into the hydrophobic core of the membrane. This mechanism explains the mode of action of a range of α-helical AMPs derived from frog skin that show selective activity against bacteria compared to eukaryotic cells. 

In previous studies, we demonstrated that a bacteriocin named Plan149a, which shows a helical secondary structure, acts by the “carpet” mechanism [57,58,59]. In the present work, the synthetized peptide P3-Ll-2085, showed significant antimicrobial activity and the same ability to inhibit the growth of *E. coli* and *S. aureus* strains. Containing 50% hydrophobic amino acid residues, this peptide had a net charge of +2 and formed an extended amphipathic α-helical structure in the presence of anionic liposomes and TFE. As it was markedly hemolytic, showing 100% hemolysis at 40 µM. P3-Ll-2085, the membrane selectivity was very low. In this regard, P2-Ll-1298 exerted higher antimicrobial activity against *E. coli* than *S aureus*. The amino acid sequence formed an extended amphipathic α-helical structure in the presence of anionic liposomes and TFE had a net charge of +2 and contained 46% hydrophobic amino acid residues. However, the membrane selectivity of P2-Ll-1298 was higher, showing low hemolytic activity at the MIC concentration.

Although other studies are necessary, according to the CD spectroscopy results and in light of the α-helix folding of P2-Ll-1298 and P3-Ll-2085 observed in the presence of negatively charged vesicle and their net charge, the antimicrobial activity of these peptides could be explained by the formation of transmembrane pores. However, the same folding was not observed for the mimicking a eukaryotic membrane (purely zwitterionic vesicles).

The mechanism of action of anionic peptides has not been elucidated. However, it is postulated that they interact with the membrane, without permeabilization, which contrasts with the mechanism shown by cationic antimicrobial peptides. Studies by Bottari et al. showed that both glutamate and aspartate form stable complexes with cations such as Zn^+2^ [32].

Information regarding anionic peptides isolated from anuran amphibians is scarce. In this regard, an anionic peptide isolated from *Bombina maxima* was identified as maximin H5, which showed activity only against one strain of *S. aureus*. In that case, the presence of Zn^+2^, and Mg^+2^ did not modify the antimicrobial activity [60].

About other animal species, two anionic peptides have recently been isolated from the hemolymph of the wax moth *Galleria mellonella* [61]. In mammals, three anionic peptides isolated from sheep lungs showed significant antimicrobial activity against *Pasteurella haemolytica*, but only in the presence of Mg^+2^ ions [62,63].

Also, the anionic antimicrobial peptide dermcidin (DCD), present only in humans, has recently been described. DCD is constitutively expressed in eccrine sweat glands and is transported through sweat to the epidermal surface [64,65]. Its mechanism of action is unknown; however, it shows potent activity against *S. aureus, E. coli* and *Candida* [66].

The anionic peptide identified in our work, P1-Ll-1577, showed considerable antimicrobial activity and a higher capacity to inhibit the growth of the *E. coli* than the *S. aureus* strain. The amino acid sequence contained 38% hydrophobic amino acid residues, had a net charge of −2, and did not adopt an amphipathic helical structure in the presence of TFE and liposomes. It showed marked membrane selectivity and non-hemolysis in the whole concentration range tested. Due to its antimicrobial activity and low hemolytic properties, P1-L1-1577 emerges as a potential candidate for the development of a new AMP, that will be further investigated not only as an antimicrobial agent, but also their therapeutics properties will be explored.

## 4. Materials and Methods

### 4.1. Collection of Amphibian Specimens

Adult specimens of *L. latrans* (Ll) (*n* = 7) were collected from the northern access to the city of Paraná (province of Entre Ríos, Argentina) during the summer months between 2006 and 2010.

### 4.2. Method for Biological Sampling. Transcutaneous Amphibian Stimulation (TAS)

Biological samples from each frog were collected, using the transcutaneous membrane stimulator method, by electrical stimulation of the granular glands [67]. All samples obtained were lyophilized and stored at −20°C.

### 4.3. Liquid Chromatography coupled to Mass Spectrometry (LC-MS-MS)

A nanoAcquity liquid chromatograph (Waters, Milford, MA, USA) coupled to an LTQOrbitrapVelos (Thermo Scientific, Waltham, MA, USA) mass spectrometer was used. Aliquots of the resuspended samples were injected for chromatographic separation in a C18 column (75 µm × 10 cm, 1.7 µm BEH column, Waters). Solvents: A: 0.1% formic acid in water; B: 0.1% formic acid in acetonitrile (ACN). The following gradient elution was used: 1 to 40% of B in 20 min, followed by 40 to 60% of B in 5 min, and a flow rate of 250 µL/min. The eluted peptides were ionized by applying an electrical potential of 2 kV with a nano-electrospray needle. The masses of the peptides were measured in Full Scan MS (Orbitrap at a resolution of 60,000 full width at half maximum or FWHM, at 400 *m*/*z*). Up to five of the most abundant peptides (minimum intensity of 3000 counts) were selected in each MS analysis to be fragmented in the High Energy Collision-Induced Dissociation (HCD) trap with helium as collision gas and normalized collision energy of 40%. Data were acquired with Thermo Xcalibur Software (v.2.1.0.1140) in raw format.

For the “de novo” analyses, Peaks Studio v5.2 software (Thermo Scientific, Waltham, MA, USA) was used (error tolerance of the peptide: 10 ppm; error tolerance of the fragment: 0.1 Da).

### 4.4. Structure Analysis

Secondary Structure Prediction

Each amino acid sequence identified by LC-MS-MS was analyzed by the PSIPRED (http://bioinf.cs.ucl.ac.uk/psipred/) and GOR V (http://gor.bb.iastate.edu/) predictive methods [68,69]. Some amino acid sequences were also analyzed through web resources for 3D structure prediction named Spark X (http://sparks.informatics.iupui.edu/) and PEP-FOLD (http://bioserv.rpbs.univ-paris-diderot.fr/cgi-bin/PEP-FOLD). This approach allowed us to obtain a predictive structural model for the most studied amino acid sequences [69,70,71,72].

### 4.5. Circular Dichroism (CD) Analyses

Far-UV CD measurements were taken on a J-810 CD spectrometer (Jasco, Tokyo, Japan) in a 0.1-cm path quartz curvet (Hellma, Mullheim, Germany) and recorded after the accumulation of five runs. CD analyses were recorded in the presence of dipalmitoylphosphatidylcholine (DPPC) and dipalmitoyl phosphatidylglycerol (DPPG) vesicles. For the preparation of small unilamellar vesicles (SUVs), the lipid dispersion in MilliQ H_2_O was sonicated, using a tip-sonicator, until the solution became transparent. In all samples, the peptide concentration was 0.2 mg/mL, and the final lipid concentration was 3 mM. To correct for background scattering caused by the vesicles, the spectrum of a single vesicle solution was subtracted from that of the peptide in the presence of vesicles. Additional spectra were obtained in the presence of trifluoroethanol (TFE) [50% TFE (*v*/*v*)] and in H_2_O. Deconvolution of CD spectra was performed using the SELCOM 3 and CONTILL methods by means of the CDPro software package (Colorado State University, Fort Collins, CO, USA) [73,74,75,76]

### 4.6. Peptide Synthesis

Peptides were synthesized by Fmoc solid-phase peptide synthesis (SPPS) as *C*-terminal amides. Couplings were performed using diisopropylethylamine (DIEA) and *N*-[(1*H*-benzotriazole-1-yl)(dimethylamino)methylene]-*N*-methylmethanaminium hexafluorophosphate *N*-oxide (HBTU)**.** Fmoc removal was achieved with 20% piperidine in DMF (*v*/*v*). Final cleavage from the resin was achieved by a mixture of TFA /H_2_O/1,2-ethanedithiol (EDT)/triisopropylsilane (TIS) (94.5:2.5:2.5:0.5) (*v*/*v*). After 3 h, the resin was filtered off, and the crude peptide was precipitated in dry cold diethyl ether, centrifuged, and washed several times with cold diethyl ether until the scavengers were removed. The product was then lyophilized twice. The synthetic peptides were purified by reverse phase HPLC using a C18 Jupiter Proteo semi-preparative column (10 µm, 90Å, 250 × 10 mm, Phenomenex, Torrance, CA, USA) with a gradient of 5–70% acetonitrile in water containing 0.1% trifluoroacetic acid. Then, these analogs were analyzed by analytical C18 RP-HPLC.

### 4.7. Minimal Inhibitory Concentration (MIC) Determination 

MIC determinations were done using the modified microtiter dilution assay, following the procedures proposed by the Hancock Laboratory for testing AMPs (http://cmdr.ubc.ca/bobh/ methods/MODIFIEDMIC.html). The target strains *Escherichia coli* ATCC 25922 and *Staphylococcus aureus* ATCC 25923 were activated by culture in Mueller-Hinton Broth (MHB) (Biokar Diagnostics, Cedex, France) for 24 h at 37 °C. An inoculum was taken and adjusted to cellular concentrations of 5 × 10^7^ CFU/mL. These inocula were used to perform the assay using diluted MHB and were incubated from 18 to 24 h at 37 °C [77,78,79,80]. The MIC was the lowest peptide concentration that inhibited the growth of each bacterial strain. All the peptides were dissolved in 36 µL of Milli-Q H_2_O with the addition of 10% acetic acid to favor their solubilization. They were then further diluted to the highest concentration of the assay (1280 µg/mL). A solution of 10 μM ZnCl_2_ and 0.14 M NaCl pH 6.7 was used as control.

### 4.8. Hemolysis Assays

The assay was performed using human red blood cells (hRBCs) and following previously described protocols [81,82].

### 4.9. Calculation of Therapeutic Index

The Therapeutic Index (TI) or specificity is defined as the relationship between the MIC and the lowest hemolytic concentration (LHC is the lowest peptide concentration that produces 100% hemolysis). When 100% of hemolysis was not detected at 320 µM, a value of 640 µM was used to calculate the TI. The index was calculated for each peptide, and bacterial strain tested. 

## Figures and Tables

**Figure 1 molecules-23-02943-f001:**
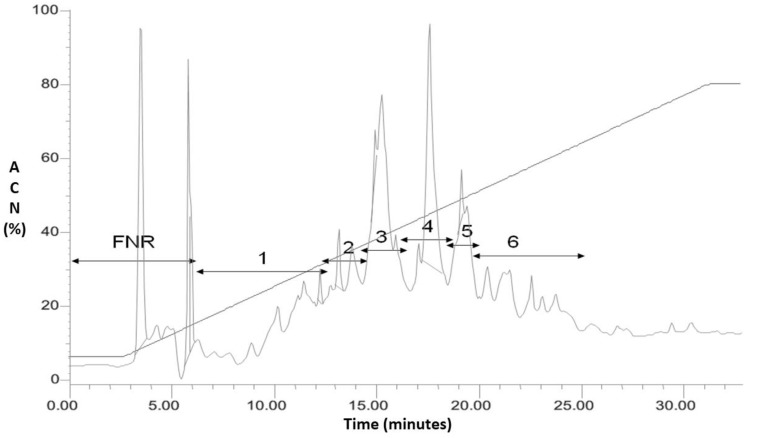
RP- HPLC of the dialyzed fraction (>1 kDa) of *L. latrans*. Detection 220 nm. FNR: fraction not retained; 1–6: fractions collected.

**Figure 2 molecules-23-02943-f002:**
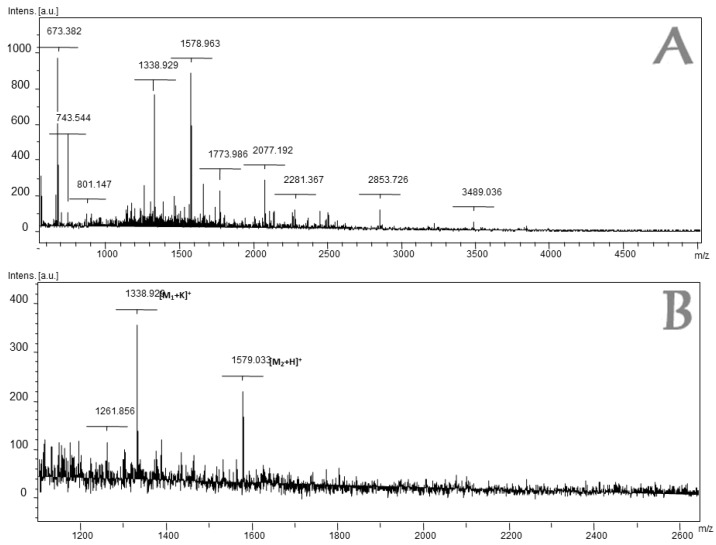
MALDI-TOF Mass Spectrum of the fraction 4 of *L. latrans*. (**A**) MS-MALDI-TOF in the range 1000–4500 Da; (**B**) Broad spectrum in the region 1100–2600 Da.

**Figure 3 molecules-23-02943-f003:**
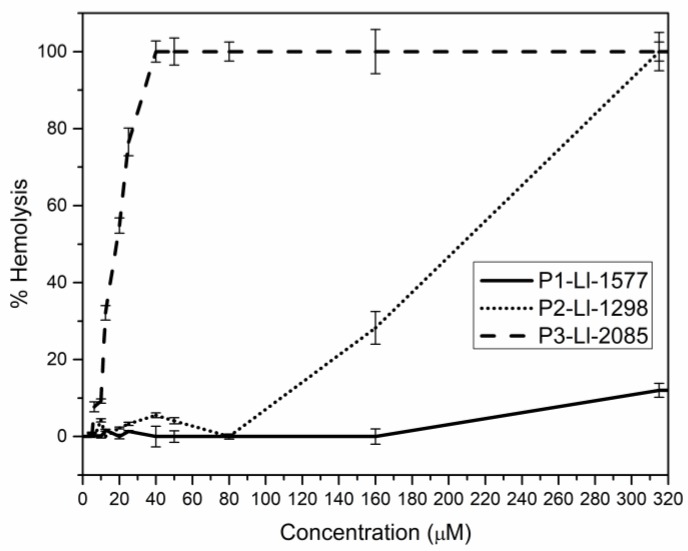
Hemolytic activity of the peptides.

**Figure 4 molecules-23-02943-f004:**
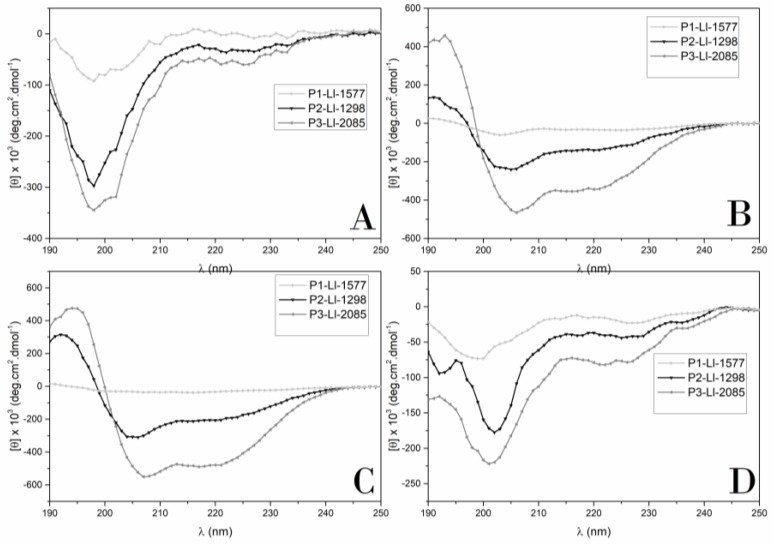
Circular dichroism spectra of synthetic analogs. (**A**) water, (**B**) TFE/ H2O (50%, *v*/*v*) (**C**) DPPG, (**D**) DPPC. Peptide concentration: 0.2 mg/mL.

**Table 1 molecules-23-02943-t001:** Sequences identified by ESI-MS-MS in fraction 4 of of *L. latrans* obtained by SE.

ID	Experimental MW	Amino Acid Sequence	Nr	Charge *	Secondary Structure Prediction		H
					PSIRED	GOR V	
**S1**	**1299.72 [M + H]^+^**	AAGKGLVSNLLEK	13	+1	Helix (K_4_-E_12_)	Helix (L_6_-L_10_)	−0.05
**S2**	**1578.71 [M + H]^+^**	DEMKLDGFNMHLE	13	−3	Coil	Coil	−0.18
**S3**	**673.382 [M + H]^+^**	GAMGKPL	7	+1	-	-	0.02
**S4**	**743.544 [M + H]^+^**	VVGDLLK	7	0	-	-	0.06
**S5**	**801.147 [M + H]^+^**	DEEAKPI	7	−2	-	-	−0.3

* Net charge at pH: 7. MW: Molecular Weight. Nr: Number of amino acid residues. H: Hydrophobicity according to the Eisenberg scale and calculated by HydroMCalc (http://www.bbcm.univ.trieste.it/~tossi/HydroCalc/HydroMCalc.html).

**Table 2 molecules-23-02943-t002:** Amino acid sequences and properties of the analogs synthesized.

ID	Amino Acid Sequence	Net Charge pH = 7	Secondary Structure Prediction		Experimental MW (*)	Rel. Hydro AA/Total AA	H
			PSIPRED	GOR V			
**P1-Ll-1577**	DEMKLDGFNMHLE-NH_2_	−2	Coil	Coil	1577.713	5/13 (38%)	−0.18
**P2-Ll-1298**	AAGKGLVSNLLEK-NH_2_	+2	Helix (K_4_-E_12_)	Helix (L_6_-L_10_)	1298.75	6/13 (46%)	−0.05
**P3-Ll-2085**	GLLDFLKAAGKGLVSNLLEK-NH_2_	+2	Helix (L_2_-E_19_)	Helix (L_3_-A_9_, L_13_-L_17_)	2085.205	10/20 (50%)	−0.01

(*) Corresponding to the Ion [M + H]^+^, determined by MALDI-TOF. H: Hydrophobicity means, according to the Eisenberg scale, calculated with HydroMCalc (http://www.bbcm.univ.trieste.it/~tossi/HydroCalc/HydroMCalc.html). Rel. AA Hydro/total AA: Ratio of hydrophobic amino acids to total amino acids. The secondary structure prediction was performed using GOR V (http://gor.bb.iastate.edu/) and PSIRED (http://bioinf.cs.ucl.ac.uk/psipred/).

**Table 3 molecules-23-02943-t003:** MIC of the analogs.

ID	MIC (µM) *E. coli* ATCC 25922	MIC (µM) *S. aureus* ATCC 25923
P1-Ll-1577 (*)	20	40.5
P2-Ll-1298	24.6	49
P3-Ll-2085	15	15

(*) Anionic peptide dissolved in 10 µM ZnCl_2_ and 0.14 M NaCl pH 6.7.

**Table 4 molecules-23-02943-t004:** Therapeutic index of synthetic peptides.

Analog Identification	MIC (μM)		MHC (μM)	Therapeutic Index (TI)	
	*E. coli*	*S. aureus*		*E. coli*	*S. aureus*
**P1-Ll-1577**	20	40.5	640	32	15.8
**P2-Ll-1298**	24.6	49	320	13	6.5
**P3-Ll-2085**	15	15	40	2.7	2.7

MHC: minimal hemolytic concentration, MIC: minimal inhibitory concentration.

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
