# Peer review of "Leptodactylus latrans Amphibian Skin Secretions as a Novel Source for the Isolation of Antibacterial Peptides"

_molecules, 2018, doi:10.3390/molecules23112943_

Round 1
Reviewer 1 Report
Reviewer’s comments (molecules-387806):
The manuscript by Siano et al. deals with the identification and structural/functional characterization of some antimicrobial peptides from the skin secretions of the frog Leptodactylus latrans. Experimentally, the authors used an approach of mass spectrometry-coupled to liquid chromatography (LC-MS-MS) to evidence a number of antimicrobial peptides (molecular masses of 1-4 kDa) from samples of skin secretions. Three peptides were selected (i.e. P1-Ll-1577, P2-Ll-1298, and P3-Ll-2085), chemically produced, and analyzed for their antibacterial potentials. They were found to inhibit the growth of Escherichia coli and Staphylococcus aureus ATCC bacterial strains. The most active peptide, P3-Ll-2085, was found to fold according to an amphipathic α-helix in the presence of trifluoroethanol (TFE) and anionic liposomes. Interestingly, addition of the ‘GLLDFLK’ motif at N-terminus of the lesser active P2-Ll-1298 significantly improved its antibacterial activity against both strains. The anionic peptide P1-Ll-1577 -which showed a significant antimicrobial potential on both E. coli and S. aureus bacteria- was highlighted as a candidate lead compound/drug for its particular membrane selectivity and apparent lack of effects on red blood cells.
In my opinion, this is an interesting work on some ‘new’ secreted antibacterials from frog’s skin. The study is (overall) technically well-done, especially the structural analyses. The manuscript is also well-written and rather clear.
However, I have some comments/concerns on this manuscript, as follows:
1. Basically, one may regret that only one Gram-negative (a strain of E. coli) and one Gram-positive (a strain of S. aureus) bacteria were used to investigate the antibacterial potential of the peptides (P1-Ll-1577, P2-Ll-1298, and P3-Ll-2085). What is the rational in the selection of these particular bacterial strains to evaluate the peptides?
2. Did the authors check whether the amino acid sequences of the characterized peptides (from 1 to 4 kDa) are not actually some proteolytic peptide fragments of some larger frog secreted molecules? Indeed, it is well-known that many enzymes are located on the skin surface to eventually degrade the peptides/proteins (the use of a mixture of protease inhibitors could thus be appropriate);
3. What is the nature of the C-termini of peptides addressed (i.e. carboxylic versus carboxylamidated)? It appears that the natural peptides showed carboxylic C-termini, contrary to the synthetic peptides with carboxylamidated C-termini;
4. Figure 1: the legends of abscissa and ordinate are missing and should be added;
5. Table 1: the values of experimental molecular masses should be homogenous (two digits after dot). The electric charge of S2 is actually ‘-3’ (instead of ‘-1’);
6. The word ‘amphibian(s)’ or ‘frog’ should appear in the manuscript’s title;
7. In the text (including Table S1), 'sequence' should be changed to 'amino acid sequence', and ‘residues’ to ‘amino acid residues’.
Author Response
The cover letter has been uploaded
RESPONSE TO REVIEWERS:
Response to Reviewer 1.
1. One of the main objectives of this work was the report of novel peptide sequences isolated from amphibian skins that exhibited significant antimicrobial activity, specifically against bacteria. One of the main disadvantages of working with natural products, particularly those obtained from wild animals, is the small amount of sample available to perform all the necessary tests since a greater quantity would imply an increase in the number of sacrificed animals. Faced with the difficulty of obtaining a considerable amount of purified material, in most cases, it is not possible to carry out the antimicrobial evaluation against a broader spectrum of microorganisms. This is why for the evaluation of the antibacterial activity of the purified fractions, two representative species of the two large groups of Gram (+) and Gram (-) bacteria were selected.
The reference strains (ATCC) that were used were Staphylococcus aureus, as representative of the Gram (+) bacteria and Escherichia coli, as representatives of the Gram (-) bacteria. The choice of these bacteria was based on the relevance as etiological agents of infectious diseases in humans: Staphylococcus aureus, is one of the main etiological agents of community-acquired infections and nosocomial infections, which pose a major public health problem worldwide causing a variety of different conditions, including wound infections, osteomyelitis, endocarditis, as well as more life-threatening diseases, such as pneumonia and bacteremia (Goudarzi et al., 2016). On the other hand, E. coli is the most common etiological agent of urinary tract infections, and some toxin-producing strains cause intestinal infections that can sometimes lead to complications such as hemolytic uremic syndrome. Also, bacteremia and meningitis are common in neonates, especially those born prematurely. (Manual MSD by Bush, LM, and Perez, MT)
To verify that the antibacterial activity of fraction 4 was due to the presence of the peptides identified and subsequently synthesized, the inhibition assays were performed using the same bacterial strains used for the evaluation of the extracts.
As it has been seen that in recent years, the prevalence of multiresistant Staphylococcus aureus has been increasing (Egea et al., 2014; Gentile et al., 2018). Also, E. coli have become increasingly resistant to conventional antibiotics. Strains resistant to multiple drugs have emerged as an important cause of urinary infection and sepsis in the community (García-Hernández, AM et al., 2011; Jiménez-Guerra, G; 2018). For this reason, the next stage of our research will contemplate the evaluation of these peptides and their analogs against a broader spectrum of bacteria, including resistant clinical isolates of E. coli and S. aureus.
2. The reviewer statement is accurate. When the extraction were performed, the pH is lowered with acetic acid in order to inactivate all enzyme activity and in addition, the temperature of the solution is lowered to 2-3 ºC with the same purpose. Followed, by freezing at -80ºC and lyophilization.
3. The reason why the peptides were amidated at the C-terminal it is because in nature the vast majority of antimicrobialpeptides (AMPs) is amidated, and in addition it has been shown that the amidation increases the antimicrobial activity, this fact is widely described in the literature: AMPs synthesized by humans and other vertebrates are amidated at the C-terminal, the amidation make these peptides able to withstand bacterial enzymes for long times (Mueller and Driscoll, 2008; Stromstedt et al.,2009).
Also, we transcribe a paragraph of the review “Animal Antimicrobial Peptides: An Overview”written by two experts in the peptide field (David Andreu and Luis Rivas): “Amidation: Perhaps the most common posttranslational modification of antimicrobial peptides, amidation occurs in a wide variety of peptides, such as melittin, cecropins, dermaseptins, PGLa, clavanin, apidaecins, diptericin, prophenin, polyphemusins or penaeidins. The process involves oxidative decarboxylation of an additional C-terminal glycine residue, in a two-step enzymatic process. Amidation prevents cleavage by carboxypeptidases and provides an extra hydrogen bond for the formation of helices“.
4. Figure 1. The legend in the abscissa and ordinate were incorporated
5. Table 1: the values of experimental molecular masses were changed to two digits after dot). The electric charge of S2 were changed to‘-3’ instead of ‘-1’;
6. The word ‘amphibian(s)’ were incorporated in the manuscript’s title
7. In the text and in Table S1, the word 'sequence' was changed to 'amino acid sequence', and the word ‘residues’ to ‘amino acid residues’.

Reviewer 2 Report
Brief description of the research:
The researchers have isolated and identified novel antibacterial peptides from skin secretions of amphibian Leptodactylus latrans. LC was used to purify the isolates and MS-MS to identify the peptide sequences. Predictive analyses were used to determine the α-helicity based on the overall charge, percentage hydrophobicity and amphiphaticity of the isolated peptides. The most cationic and hydrophobic peptide was analysed for other homologous sequences through AMP database which allowed for the rational design of a hybrid AMP. Three peptide sequences were synthesised using SPPS followed by testing of their antimicrobial and hemolytic activity and structural elucidation using CD in four 4 distinct environments: water, TFE/water, anionic and zwitterionic vesicles.
Novelty:
While two of the three peptides synthesised and tested had adopted an amphipathic α-helical structure and with a net positive charge inhibited the growth of Escherichia coli and Staphylococcus aureus (representing gram-negative and gram-positive) as expected, the third peptide which also exhibited antimicrobial activity corresponded to an unstructured anionic peptide. The anionic peptide showed minimal hemolytic activity while the two amphipathic α-helical peptides showed significant hemolysis which is consistent with the literature. While there are innumerable examples of cationic α-helical AMPs the identification of an unstructured anionic peptide is of great interest as it presents increased selectivity towards the microbial membranes thereby resulting in greater therapeutic index.
Underlying mechanism:
Detailed discussion for the mode of action for the amphipathic α-helical cationic peptides is presented including the barrel-stave model implying the formation of transmembrane pores/channels and a carpet model implying membrane permeabilization via detergent like activity. However, there is not much discussion on the unstructured anionic peptide. Even though the authors allude to the fact that mechanism of action for anionic peptides has not yet been elucidated and the presence of cations (Zn+2, Na+, Mg+2) is crucial for activity, mentioning that both glutamate and aspartate form stable complexes with cations, further explanation predicting its mode of action would have been appreciated.
Author Response
Response to Reviewer 2.
The authors of the present work agree with the comments of this reviewer. Especially regarding the mode of action of the anionic peptide. We consider that discuss more about the mechanism of action without experimental confirmation could result in a not so exact interpretation of what is truly happening. This is why; the next step of our research will be focused on understanding the mechanism of action of this peptide and the influence of different cations in the biological activity.
Reviewer 3 Report
Interesting results are reported, and the conclusions and major findings are useful to the community. However there are some things I think the authors should change to improve on their manuscript.
Table 1 should be improved, spaces are not well distributed and some sentences are separated (at PSIRED e.i. (K4-E12)
Table 2. Authors stated that for the synthetic peptides the experimental MW Corresponds to the Ion [M]+, but actually it is the Ion [M+H]+
For Figure 1 (page 5). Please include the axis titles
For the three synthetic peptides, it should be reported the chromatographic profile and the MALDI-TOF MS spectra.
At materials and methods section (peptide synthesis), it is not clear if synthetic molecules were purified or not.
At materials and methods section, (Determination of Minimal Inhibitory Concentration (MIC) and Hemolysis Assays), It should be stated if the solution of 10 μM ZnCl2 and 0.14 M NaCl pH 6.7 was used as control. If so, please indicated at MICs table and Figure 3
Through the manuscript, some references are not separated from the text, for example Page 4, lines 7 to 10.
At Page 3, Ln 18 an space should be introduced: ”….positive net charge(that varies from +2 to 9)…”
Author Response
1. Table 1 was improved
2. Table [M]+ was changed to [M+H]+
3. The axis were added in Figure 1
4. For the the three synthetic peptides the chromatographic profile was added in supplementary
information (Figure S).
5. At material and methods section, in peptide synthesis a paragraph was added regarding the
HPLC purification of the peptides.
6. In Material and Methods section, it was added that the solution of 10 μM ZnCl2 and 0.14 M NaCl
pH 6.7 was used as control.
7. The references were separated from the text
8. At Page 3, Ln 18 a space was introduced